# Nighttime Cloud Cover Estimation Method at the Saishiteng 3850 m Site

Baoquan Gao [1,2,3],* , Yiding Ping [1,3] , Yao Lu [1,3] and Chen Zhang [1,2,3]

1 Purple Mountain Observatory, Chinese Academy of Sciences, Nanjing 210023, China
2 School of Astronomy and Space Science, University of Science and Technology of China, Heifei 230026, China
3 Key Laboratory for Space Objects and Debris Observation, Chinese Academy of Sciences, Nanjing 210023, China
* Correspondence: bqgao@pmo.ac.cn

**Abstract:** Cloud cover is critical for astronomical sites because it can be used to assess the observability of the local sky and further the fractional photometric time. For cloud monitoring in site-testing campaigns with all-sky cameras, previous studies have mainly focused on moonless images, while the automatic processing methods for moonlight images are explored quite few. This paper proposes an automatic estimation method for cloud cover, which takes all cases of nighttime gray-scale all-sky images into account. For moonless images, the efficient Otsu algorithm is directly used to detect clouds. For moonlight images, they are transformed into cloud feature image using a colorization procedure, and then the Otsu algorithm is used to distinguish cloud pixels from sky pixels on the cloud feature image. The reliability of this method was evaluated on manually labeled images. The results show that the cloud cover error of this method is less than 9% in all scenarios. The fractional photometric time derived from this method is basically consistent with the published result of the Lenghu site.

**Keywords:** astronomical site testing; cloud detection; cloud cover; all-sky camera; image processing





## 1. Introduction

The Tibetan Plateau, the highest plateau on Earth with an average elevation of over 4000 m, can potentially provide excellent astronomical sites. The latest research shows that Saishiteng Mountain, located in the Qaidam Basin of the Tibetan Plateau, has a highly arid climate, a clear night sky, and good atmospheric transparency due to its unique geographical conditions ([1]). Furthermore, the stable airflow and small diurnal temperature variation on the peaks around 4000 m high result in excellent astronomical seeing with a median of 0.75 arcseconds. Because of these advantages, the mountains near Lenghu Town in Qinghai Province have become an ideal astronomical site. So far, several Chinese astronomical science projects have been confirmed to land on the site, including the Multi-Application Survey Telescope Array (MASTA), a project of the Purple Mountain Observatory of the Chinese Academy of Sciences, which is under construction and scheduled to run in 2023.

A peak(called the 3850 m site) on Saishiteng Mountain was chosen for the MASTA project, and a site-testing campaign has been conducted since the end of December 2019. The 3850 m site (longitude = 93°53′ E, latitude = 38°35′ N) is about 2 km south of the current Lenghu site. To validate the site-testing results of the 3850 m site independently and also to preserve data for monitoring the observational condition in the long term, we deployed a suite of site-testing instruments, including a weather station, all-sky cameras, Sky Quality Meter (SQM), differential image motion monitor (DIMM), and three Cyclope Seeing Monitors.

Another purpose of deploying these devices is to improve the guidance and optimization of MASTA's observational schedules in the future. MASTA is a large-scale optical

survey project, covering a wide range of scientific goals, including Near-Earth Objects, transient events, variable stars, etc. MASTA will be deployed at the 3850 m site on Saishiteng Mountain, comprised of 20 optical telescopes. Each telescope has an aperture of 710 mm, and its designed optical field of view is $6° \times 6°$. Estimating the nighttime cloud cover can not only provide us a with general understanding of the observability and fractional photometric time at the 3850 m site, but also allow us to run MASTA more efficiently in the future. As a result, estimating cloud coverage in the nighttime on both moonless and moonlight images is necessary.

All-sky cameras are commonly used to monitor the local sky environment above astronomical site [2–7]. There are several approaches used to measure cloud cover in all-sky nighttime images.

For gray-scale nighttime images, Yin et al. used astronomical photometry to detect stars, using the percentage of detected stars to estimate the mean cloud cover for moonless images and the formula of the CIE (Commission International de l'Eclairage) for moonlight images [8]. Shamir et al. created a set of reference clear sky images and used them to compare with images taken at the same sidereal time to obtain cloud cover. This method needs a sufficient number of clear night images [9]. Gacal et al. proposed a single fixed threshold method to segment nighttime cloud images, but it is difficult to accurately segment cloud pixels due to complex weather conditions [10]. Jankowsky et al. analyzed data from the Automatic Telescope for Optical Monitoring system using astronomical photometry, then compared it with theoretical nights to estimate cloud coverage [11].

For color nighttime images, traditional methods generally use different color spaces and specific color ratios as a distinguishing factor between clouds and clear sky. Skidmore et al. assessed the clouds for Thirty Meter Telescope candidate sites using a manual method based on blue and red colors [12]. Dev et al. proposed and achieved good results with a super-pixel algorithm for detecting clouds in nighttime images [13]. Kemeal et al. proposed a robust hybrid method to measure the clouds [14].

However, so far, the methods used for automatically processing the moonlight images are quite few, particularly for gray-scale images. On moonlight sky, the area away from the Moon is also crucial, since some of these skies are relatively clear and can meet the observing requirements completely. Making use of these clear skies can significantly improve the overall observation efficiency and utilization of the MASTA system.

In this paper, we present a cloud segmentation approach for nighttime gray-scale images to estimate cloud cover. The method is based on the Otsu dynamic threshold algorithm and image colorization, and can successfully separate cloud pixels from the clear sky in moonlight and moonless images. Our technique is remarkable in that it can identify clouds in gray-scale moonlight images. The all-sky camera system and data preprocessing are briefly discussed in Section 2. Section 3 describes the cloud segmentation approach for all-sky nighttime images. In Section 4, we conduct three sets of tests to validate our approach. Finally, the conclusions are given in Section 5.

## 2. Instrument and Data

### 2.1. Camera System

We installed a Starlight Xpress Oculus all-sky camera on the top of a 12-meter-high tower, as shown in Figure 1. To prevent damage by wind, rain, snow, and fog, the camera is protected by a clear glass cover. The camera system is kept warm by connecting the internal heater via a 12v connector to remove frost and snow from the glass dome and moisture from the lens. AllSkeEye software (https://allskeye.com/ (accessed on 30 September 2022)) is used to control thw all-sky camera automatic observation, which can be performed continuously for 24 h without artificial management. During dusk and dawn, the camera continuously captures one gray-scale all-sky image per 5 min, regardless of the weather conditions, and the AllSkeEye software controls time exposure automatically. Local storage is used to save images in FITS and JPEG formats. Table 1 summarizes some critical specifications of the camera system.

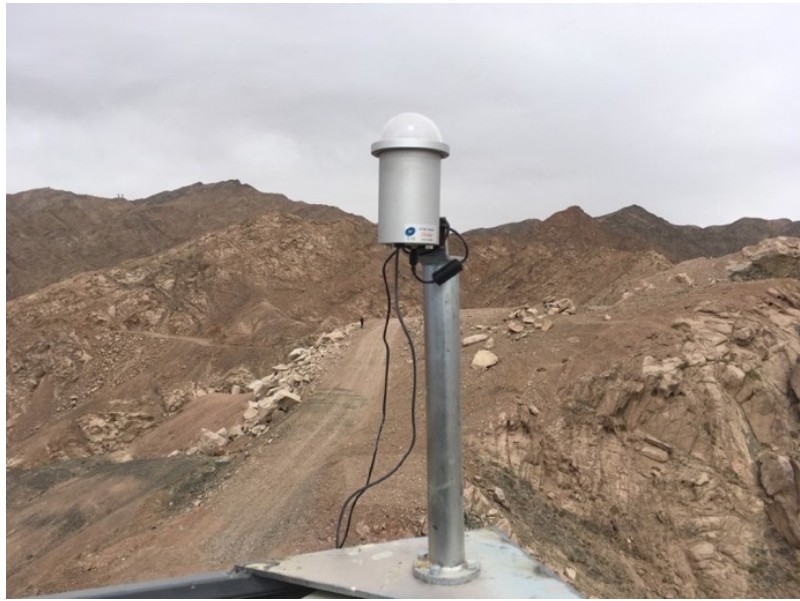

**Figure 1.** The appearance of the all-sky camera.

**Table 1.** The camera system specification.

| Parameters | Values |
|---|---|
| CCD type | ICX267AL Sony CCD |
| CCD detector | 1392 × 1040 |
| Spectral Response | QE max at 520 nm (50%), 30% at 420 nm and 670 nm |
| Anti-blooming | greater than 1000× |
| Full-well capacity | greater than 15,000 e$^-$ |
| Lens | 1.5 mm F/2 180 degree fish eye |
| Exposure time | 2 ms∼90 s |
| Dome | polycarbonate hemisphere |
| Color representation | Monochrome |

*2.2. Data Preparation*

Since we are only interested in nighttime all-sky images between astronomical evening twilight and astronomical twilight, we compute the altitude of the Moon at a specific time given the longitude and latitude of the 3850 m site to filter the data set. Our image data does not include complex air-condition images, such as cloud images under rain and dew. The sky plane is projected as a circular region with a diameter of about 960 pixels. The pixels that are not sampling the sky must be removed, such as buildings, mountain ridges, and ground (as shown in Figure 2a). The masking technique is used to produce region of interest images (ROI) by setting the corresponding pixel values to zero. Each image is resized to 960 × 960 pixels (as illustrated in Figure 2b).

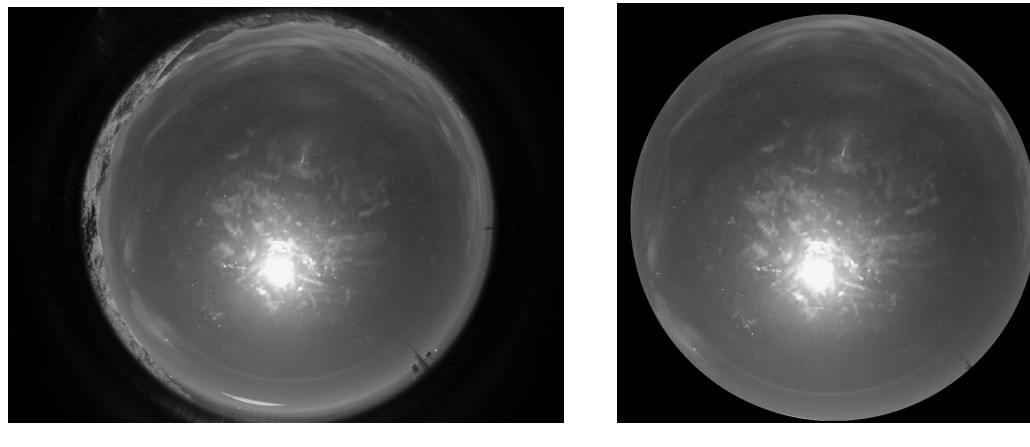

(**a**) Raw image                    (**b**) Processed image

**Figure 2.** Preparation of nighttime image. (**a**) Raw image of the local sky, with non-relevant sections and a circular projection of the sky. (**b**) Image has been cropped and masked (buildings and ridges).

## 3. Method

Clouds may appear brighter or darker than the clear night sky depending on the Moon's illumination conditions. The comparison between a moonlight image (MI) and a moonless image (NMI) is shown in Figure 3. When the image contains moonlight (MI), the area of clouds is substantially brighter than the clear sky, and the Moon area has brightness characteristics similar to those produced by clouds. In a moonless image (NMI), on the other hand, clouds are darker than the clear sky, resulting in a sharp peak at the darkest end of the histogram of the pixel values. Thus, these two types of images show significantly different characteristics, which requires us to deal with them separately.

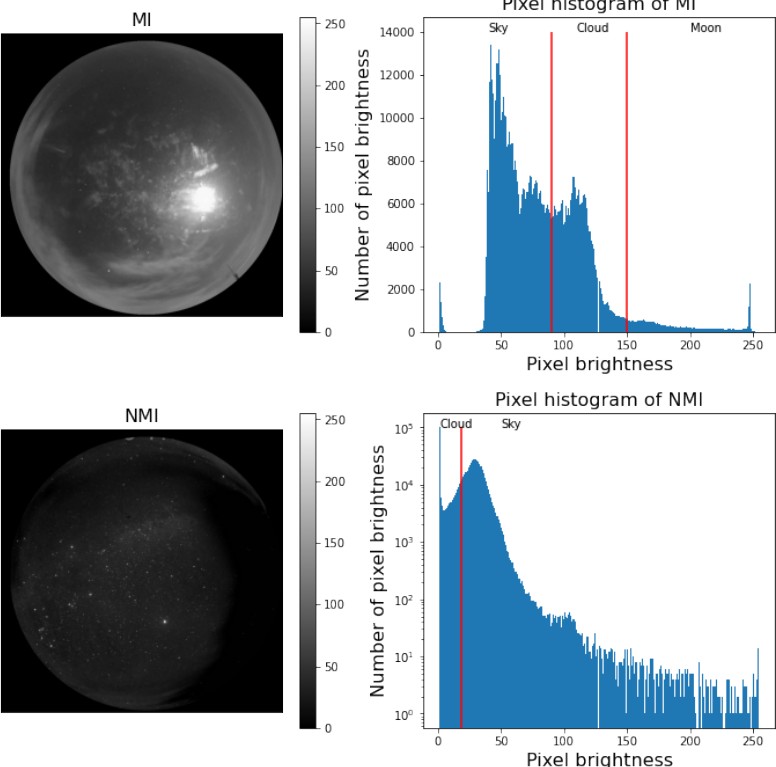

**Figure 3.** Two examples of all-sky images. Clouds (white) appear brighter than the clear sky in MI (**upper panel**) and clouds appear darker than the clear sky in NMI (**lower panel**). The histogram is presented in the right panels. The red line indicates the threshold value, which is obtained by our method in this paper.

Threshold segmentation is a traditional image segmentation method, which is the most basic and widely used segmentation technique in image segmentation. Due to the complex light conditions at night, the pixel values for nighttime images with cloud and sky backgrounds are extremely low and difficult to distinguish. In comparison to the fixed threshold, the Otsu method [15] has a stronger reliance on nighttime environmental conditions than the fixed threshold method, and can deliver the desired results.

### 3.1. Method of NMI

In NMIs, the brightness of starlight will be reduced or completely blocked in the area where clouds are present. As a result, the cloud pixel values are significantly lower than the rest of the image. In the lower-right part of Figure 3, it can be seen that the values of cloud pixels are distributed in a very narrow range, while the values of the remaining pixels follow an almost normal distribution. NMI contains two classes of pixels (cloud and sky), corresponding to two peaks on its histogram. The left peak is the distribution of brightness pixels with a value of 0 (those pixels are completely cloud-covered). Such a bimodal distribution is very suitable for any threshold-based image segmentation method. In our approach, we apply the Otsu method, which is a classic automatic image threshold method for non-parameter and unsupervised segmentation, as our solution for NMIs.

The core idea of the Otsu method is to separate the image into foreground and background using an optimal threshold, which is the outcome of minimizing the weighted variance of the intensities of foreground and background pixels. It is a statistical approach widely used for automatic segmentation.

Let $t$ denote a threshold with an initial value of any value between 0 and 255, which separates the image into two classes. Then, the intra-class variance is defined as the weighted sum of the two classes' variances:

$$\sigma_w^2(t) = \omega_0(t)\sigma_0^2(t) + \omega_1(t)\sigma_1^2(t) \tag{1}$$

where $\omega_0(t)$ and $\omega_1(t)$ are the probabilities of the two classes. $\sigma_0^2(t)$ and $\sigma_1^2(t)$ are the variances of the two classes respectively. Stepping through $t$ exhaustively one can find the value that minimizes the intra-class variance, which is the desired threshold.

On NMI, we used the Otsu method to segment clouds from sky pixels; an example is shown in Figure 4. We can see that cloud distribution in the left panel is almost consistent with the right panel, which proves that the Otsu method can effectively segment clouds in NMIs.

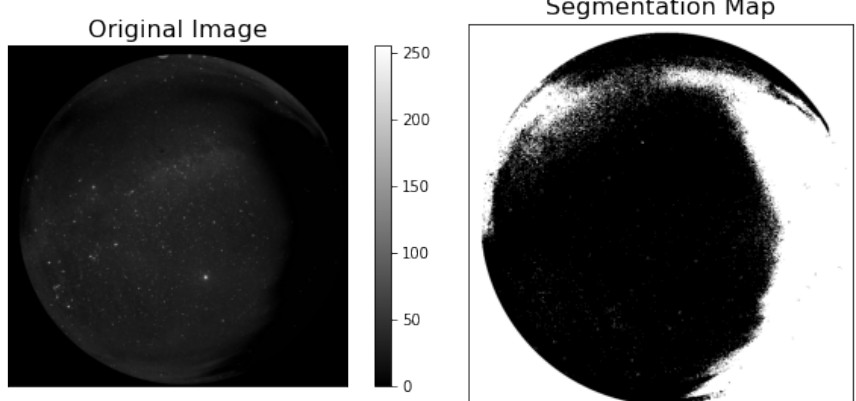

**Figure 4.** An example of the derived cloud segmentation map based on the Otsu method. Original moonless image (**left**); the black color represents the regions covered by clouds. Segmentation map (**right**); the regions covered by clouds are indicated in white color.

### 3.2. Method of MIs

In the case of MIs, on the other hand, the sky is dominated by moonlight and the clouds are illuminated by the Moon. Therefore, the clouds in MIs are brighter than the clear-sky areas. The Moon area with higher brightness values is detected as the clouds in the upper-left part of Figure 3; it will be incorrectly detected as clouds. As a result, we need to mask the Moon area of MIs to mitigate its effects.

#### 3.2.1. Moon Location

Given the geographical latitude and longitude of the 3850 m site and observation time, the horizon coordinates of Moon in any MI can be calculated using the Astropy software package [16]. The Cartesian coordinates of the position $(x,y)$ of the Moon on MIs are determined by the following equations:

$$x = x_0 - Rsin\theta \tag{2}$$

$$y = y_0 + Rcos\theta \tag{3}$$

where $(x_0,y_0)$ are the zenith coordinates in MI, $R$ is the polar distance from the center of the MIs, and $\theta$ is the polar angle. Because of the lens distortion, the relationship between $R$ and $\theta$ is nonlinear. We model this relationship with a cubic polynomial by fitting pairs of $R$ and $\theta$:

$$R = a \times \theta + b \times \theta^2 + c \times \theta^3 \tag{4}$$

According to the Moon's Cartesian coordinates, a circular mask of the Moon can be created automatically. It should be noted that since astronomical observations are not interested in the Moon region, we use a circular mask with a fixed radius regardless of the Moon phase. Figure 5 shows a masked MI and pixel histogram. We can see that the image histogram has an approximate single-peaked distribution with little difference in contrast between clouds and clear sky. However, the masked MI has some unmasked regions affected by moonlight (the right side of the histogram).

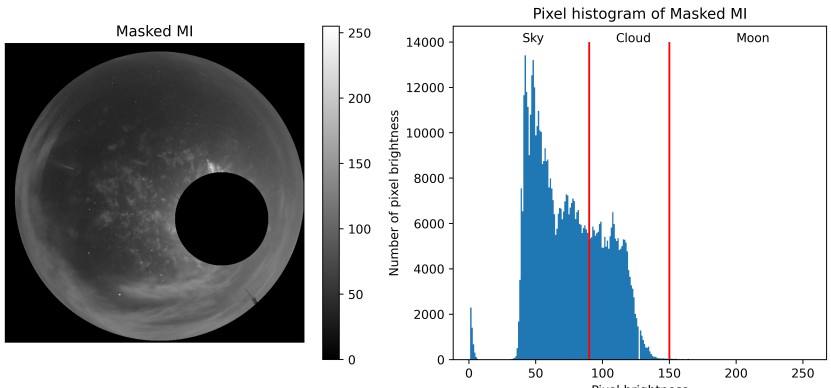

**Figure 5.** (**Left**) A masked moonlight image. (**Right**) Image histogram, the left vertical line indicates the threshold value, which is obtained by our method, and the right vertical line is the gray value 150.

In Section 3.1, the Otsu method has been shown to achieve good cloud segmentation results in NMIs. Figure 6 shows the segmentation image obtained when applying the Otsu method to masked MI. It is clear that all of the clouds remain undetected by this procedure. This experiment illustrates that applying the standard Otsu method does not provide the optimal segmentation of clouds and clear regions. We need a new methodology to process MIs for better performance.

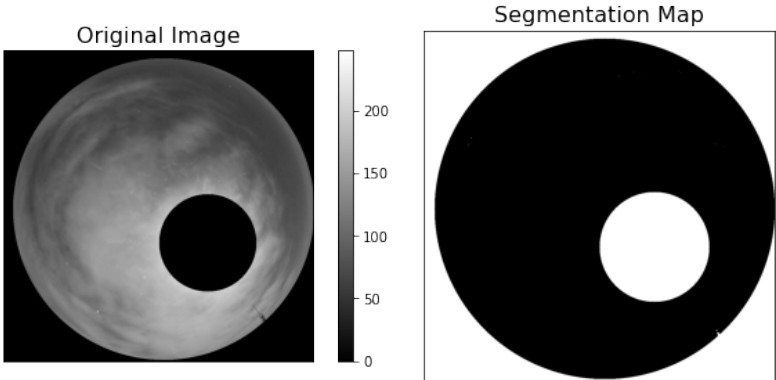

**Figure 6.** An example of the derived cloud segmentation map based on the Otsu method. The black color represents the regions clear of clouds and masked Moon, while the white color corresponds to regions covered by clouds.

### 3.2.2. Cloud Feature Image

We adopted a colorization procedure to generate a good proxy of the cloud cover following the techniques described in [17,18]. First, an RGB image was created using Equations (5)–(7). The maximum and minimum values between the three channels of the RGB image are then estimated for each pixel (Equations (8) and (9)). Finally, we define a cloud feature image that will be demonstrated to accurately trace the location of the clouds as:

$$R = 1.164 \times (MI - 16) + 1.596 \times (MI - 128) \tag{5}$$

$$G = 1.164 \times (MI - 16) - 0.392 \times (MI - 128) - 0.813 \times (MI - 128) \tag{6}$$

$$B = 1.164 \times (MI - 16) + 2.017 \times (MI - 128) \tag{7}$$

$$Max = max(R, G, B) \tag{8}$$

$$Min = min(R, G, B) \tag{9}$$

$$f = \frac{Max - Min}{Max} \tag{10}$$

The complete algorithm for colorization of gray-scale MIs is summarized as follows in Algorithm 1:

---
**Algorithm 1** colorization of MIs

---
    **Data:** gray-scale MI
    **Result:** cloud feature image
  1  Convert gray image into three layers by repeating gray image for each layer;
  2  Convert to RGB color space;
  3  Calculate the maximum and minimum value;
  4  Get cloud feature image;

---

This is the image adopted to apply the Otsu method to estimate the cloud cover in MIs. Figure 7 shows the cloud segmentation map, which proves that the Otsu method can successfully segment clouds after using our method.

The essence of our algorithm is a histogram equalization technique that increases the contrast between the cloud pixels and the clear sky pixels in MI, resulting in a more easily segmented cloud image (cloud feature image). Figure 8 represents the relationship between the pixel values in the two images. The fitting function is as follows:

$$F = \begin{cases} 0.00072 \times MI + 1.889, & \text{if } (MI < 83) \\ -0.042 \times MI + 5.3985, & \text{if } (83 \leq MI < 127) \\ 0.3 \log(0.5MI - 60.7) - 0.364, & \text{if } (127 \leq MI < 160) \\ MI^{0.42} - 0.735, & \text{if } (160 \leq MI \leq 255) \end{cases} \tag{11}$$

where $F$ is the pixel values of the cloud feature image, $MI$ is the pixel values of the $MI$. Based on this equation, the $MI$ can be quickly converted to cloud feature images.

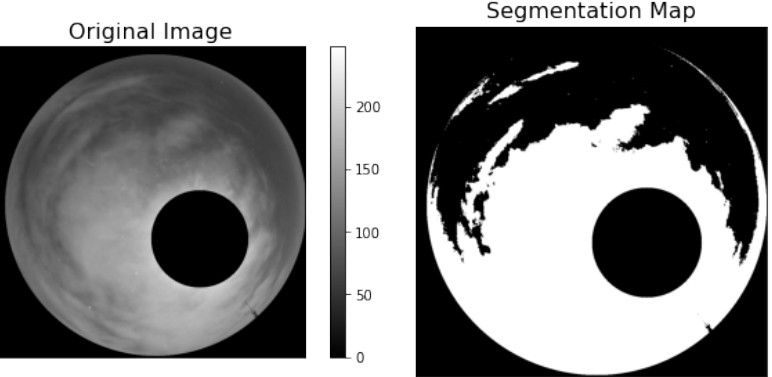

**Figure 7.** An example of the derived cloud segmentation map based on the adopted procedure described in Section 3.2.2 for the MIs. The black color represents the regions clear of clouds and masked Moon, while the white color corresponds to the regions covered by clouds.

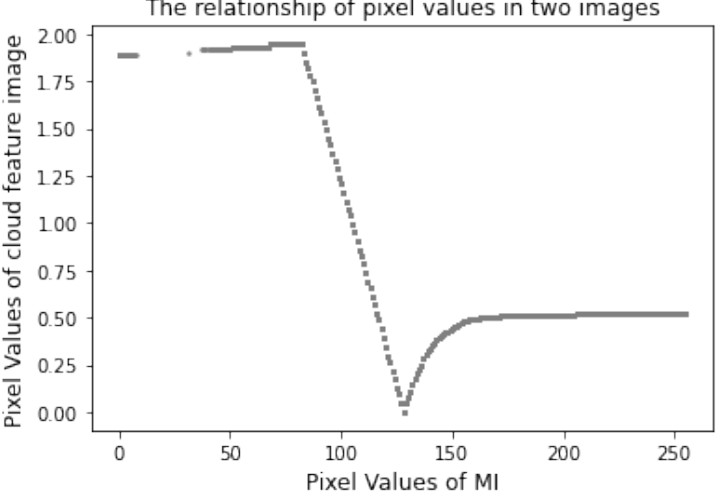

**Figure 8.** Mapping of pixel values between two images. The horizontal axis represents the pixel values of the MI, and the vertical axis represents the pixel values of the cloud feature image.

*3.3. Cloud Cover Ratio*

Following the application of the two described procedures for NMIs (direct Otsu method) and MIs (Otsu method on the cloud feature image), a segmentation map for each exposure image (with 0 indicating clear sky and 1 indicating the presence of clouds) is obtained. This segmentation map is used to derive the Cloud Cover Ratio [19]:

$$CCR = \frac{N_{cloud}}{N_{sky} + N_{cloud}} \times 100\% \tag{12}$$

where $N_{cloud}$ is the number of cloud pixels and $N_{sky}$ represents the number of sky pixels in an all-sky image (masked pixels are not included in MIs). According to the cloud segmentation results from the method mentioned above, the cloud fraction can be estimated.

## 4. Result and Discussion

Figure 9 shows four examples of various NMIs with different sky conditions: a fully clear sky (top panels), a sky with just a few clouds (second row of panels), a partially cloudy sky (third row of panels), and totally overcast (bottom panels). A version of the same image adopting an inverse color map is shown in the second column. Finally, the resulting segmentation maps once applied the procedures described in Sec NN are included in the third column. These results indicate the ability of the Otsu method to detect both thick and thin clouds in NMIs. With an accuracy better than 4% compared to manually labeled results (see below for details of the labeling method), it is clear that the segmentation precisely reflects the real situation no matter what the sky conditions are, which makes the Otsu method an excellent strategy for NMIs. Although dust cannot be seen in NMIs, it does exist due to long-term outdoor operation. The dust in the center of the NMI images is not detected as clouds, demonstrating that it does not influence our cloud cover estimation.

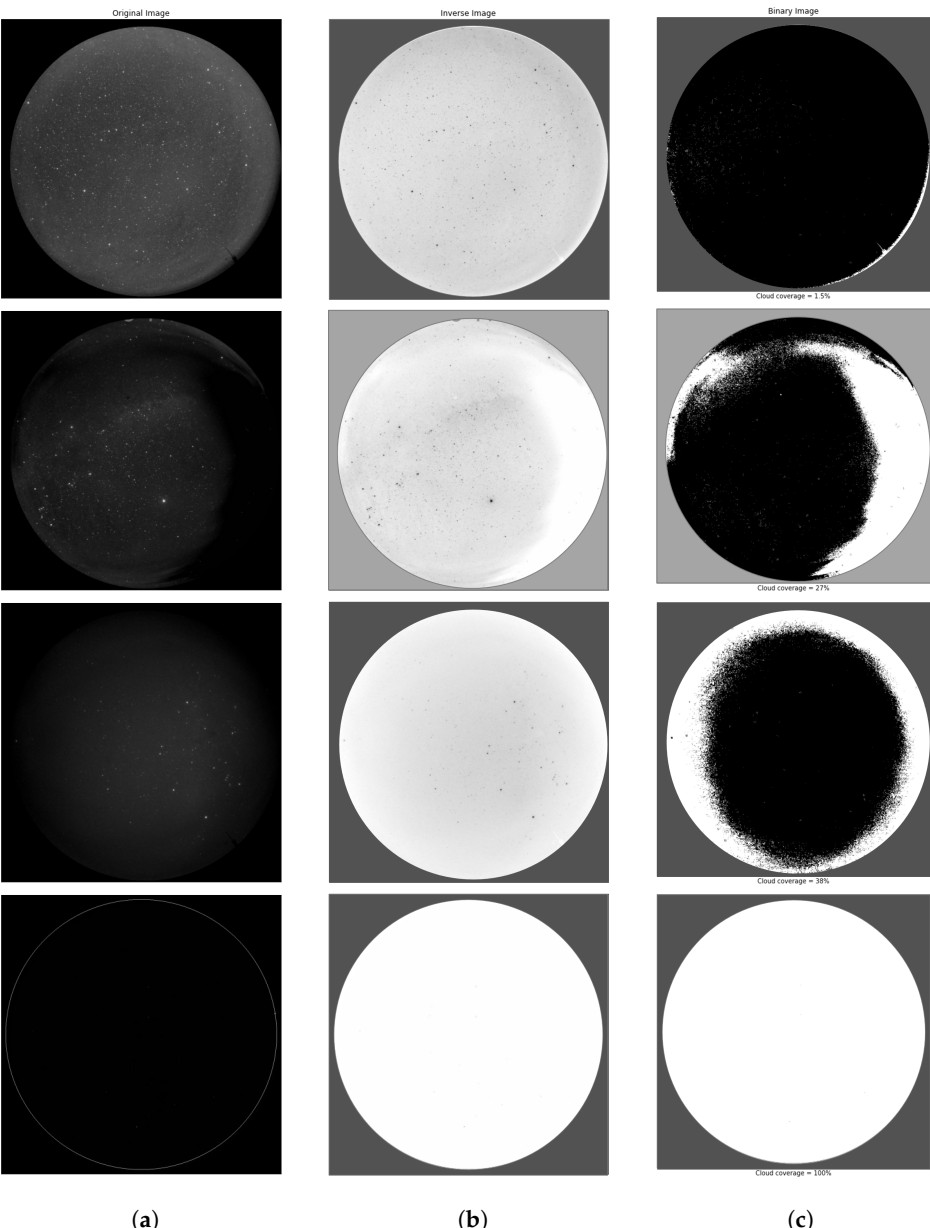

**Figure 9.** Four different sky conditions for NMIs. (**a**) Original NMIs. (**b**) Inverted color map NMIs. (**c**) Segmentation map, where the black color represents the regions clear of clouds and the white color corresponds to regions covered by clouds.

Figure 10 is similar to Figure 9, as it includes four examples of different MIs corresponding to the same cloud coverage as in that figure. Although there are still flaws in the masked Moon area, the results are remarkably close to the cloud distribution of the corresponding MIs. Very thin clouds are not detected. There are still errors in the detection of the masked Moon area. We estimate the error rates in the next part, which is used to revise the clear sky percentage specifically for the all-sky system and site.

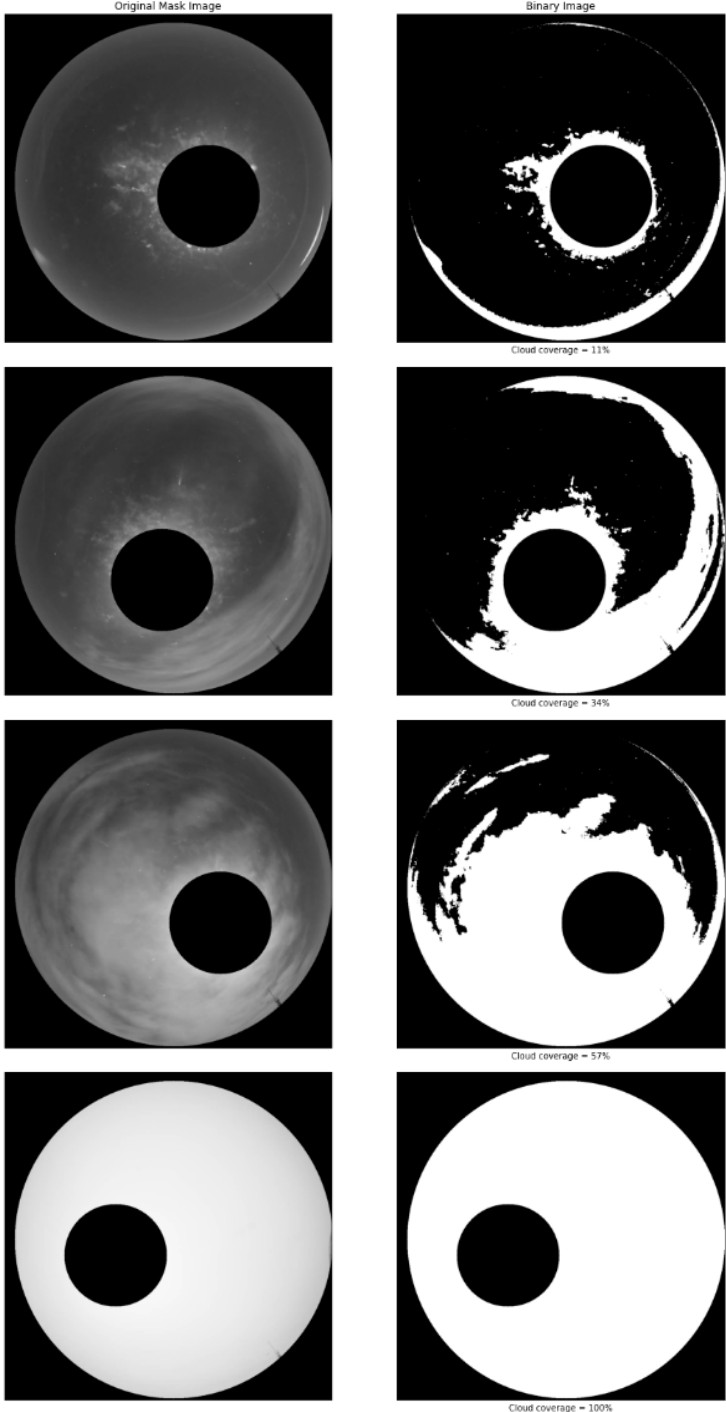

**Figure 10.** Four different sky conditions for MIs. (**Left**) Masked MIs. (**right**) Segmentation map, where the black color represents the regions clear of clouds and the white color corresponds to regions covered by clouds.

To further verify the accuracy and practicability of our method, we randomly selected 400 images as our data set, including 200 NMIs and 200 MIs. First, we classify the images into four categories based on the sky conditions: fully clear sky, few clouds, partially cloudy, and totally overcast. Second, by using the two described procedures for NMIs and MIs, the images are transformed into segmentation maps, and then the cloud cover ratio is calculated using formula (12), which is called the current analysis results. Finally, we use LabelMe software [20] to label clouds in these images based on the brightness characteristics described in Section 3.1 to obtain the labeled cloud maps. Manual labeling also means that human subjective uncertainty has an impact on the manually labeled results, particularly in the Moon area. We calculate the cloud cover ratio of labeled cloud maps and then set them as manually labeled results.

The comparison of the current analysis and manually labeled results is shown in Table 2. We should note that the cloud cover ratio of MIs does not include masked pixels. As demonstrated in the table, when the sky condition is clear sky or presents a few clouds, the Moon area has the most errors in cloud segmentation in MIs, and roughly 9% of the sky background pixels are mistakenly segmented into cloud pixels. As the percentage of the cloud cover rises, the current analysis results become more accurate, and these are more accurate when there are few clouds in NMIs, whereas the inaccuracy is higher when the sky is overcast. These results demonstrate that the approach proposed in this paper can properly segment clouds in all-sky nighttime images and that it has practical implications for future automated cloud coverage studies.

**Table 2.** Comparison results of current analysis and manually labeled results. The average error is defined as the average of the analysis results minus manually labeled results of multiple images in each category.

| Image Category | Average Error | | | |
| :---: | :---: | :---: | :---: | :---: |
| | **Clear Sky** | **Few Cloudy** | **Partly Cloudy** | **Overcast** |
| MIs | 8.9% | 6.4% | 4.1% | 1.5% |
| NMIs | 3.9% | 0.8 % | 2.8% | 7.2% |

Our method is also used to analyze the fraction of clear sky at the 3850 m site. Figure 11 shows the cloud cover statistics of the 3850 m site. It is found that 81% of the images have a cloud coverage lower than 50% of the full sky; a cloud coverage lower than 33% of the full sky is 60% per year. In order to compare with the Lenghu site, we adopt the definition of photometric (cloud coverage $\leq$ 15%), Spectroscopic (12.5% < cloud coverage < 50%), and unusable (cloud coverage $\geq$ 50%) in Ref. [21], which was used to analyze the fractional photometric time of the Thirty Meter Telescope and European Extremely Large Telescope sites [22,23]. The result shows that the fractional photometric time is 71% at the 3850 m site, which is basically identical to the results of the Lenghu site ([1]).

One of the issues we need to highlight in this study is the influence of the Moon. MIs will have a blooming effect, especially when taken under clear sky conditions. Figure 12 shows a MI taken under clear sky conditions and the corresponding binary image. Because the pixel values of the Moon area are mostly close to cloud pixels, and thus would be considered cloud pixels, it was extremely difficult to distinguish the pixels in MI. Limited by the gray-scale format, color information cannot be used to characterize these pixels. In order to improve the effectiveness of this type of cloud detection method, more research is needed to differentiate and characterize the pixels.

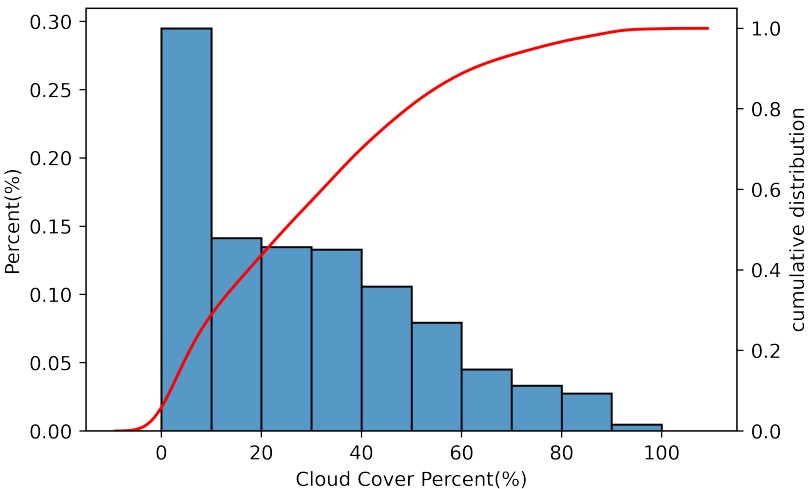

**Figure 11.** Distribution of cloud fraction.

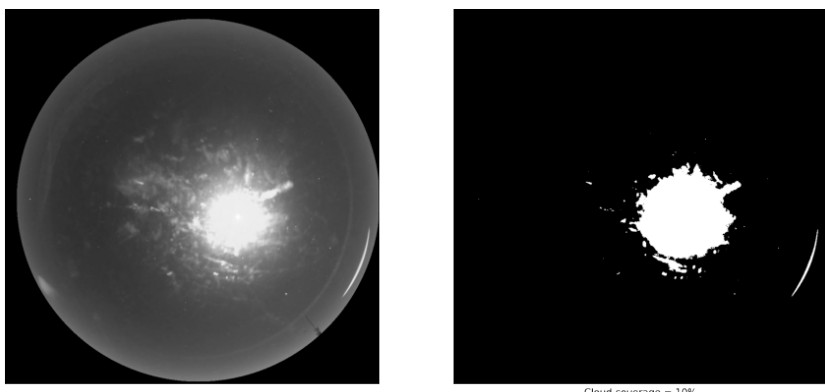

**Figure 12.** (**Left**) MI taken under clear sky conditions. (**Right**) The binary image, where cloud cover of the Moon area is 10%.

## 5. Conclusions

Cloud cover plays a critical role in the site quality of astronomical sites and optical telescopes. In this paper, a cloud segmentation method is proposed to segment cloud pixels from moonless images (NMIs) and moonlight images (MIs). The Otsu method is applied to segment clouds of NMIs. The MIs are processed by a colorization technique to extract the enhanced cloud feature image, and then the Otsu method is applied to segment clouds. The four different types of cloud cover images (clear sky, low level of cloud coverage, partially cloudy, and overcast) are chosen to validate the method. The results of NMIs and MIs show that our method can effectively segment clouds in nighttime all-sky images and automatically estimate cloud cover ratio. A comparison with manually labeled results further proves its accuracy and practicability.

The determination of cloud pixels in MIs remains flawed in areas with strong illumination of the Moon. In our next work, we will define distinct sky zones and develop observation strategies based on cloud coverage to guide the astronomical observation process of the Multi-Application Survey Telescope Array (MASTA).

**Author Contributions:** Conceptualization, B.G.; Data curation, B.G.; Formal analysis, B.G.; Validation, Y.L.; Writing—original draft, B.G.; Writing—review and editing, B.G., Y.P. and C.Z. All authors have read and agreed to the published version of the manuscript.

**Funding:** Our work was funded by the National Natural Science Foundation of China (Grant Nos. 12173095, 1167070).

**Acknowledgments:** The author would like to thank the MASTA project.

**Conflicts of Interest:** The authors declare no conflict of interest.

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
