# Peer review of "Nighttime Cloud Cover Estimation Method at the Saishiteng 3850 m Site"

_universe, doi:10.3390/universe8100538_

Round 1

Reviewer 1 Report (Previous Reviewer 1)

Since the authors have followed up on all my concerns and suggestions, and have substantially improved the manuscript, I cannot see any serious barriers to publication. Although some typographical errors remain in the manuscript, these can be corrected at the printing stage.

Author Response

Comments from the author:
We would like to thank you for your careful reading, helpful comments, and constructive suggestions, which have significantly improved the presentation of our manuscript.

Comments:

Since the authors have followed up on all my concerns and suggestions, and have substantially improved the manuscript, I cannot see any serious barriers to publication. Although some typographical errors remain in the manuscript, these can be corrected at the printing stage.

Response:
Thank you for your comments. I have corrected spelling errors in the manuscript.

Yours sincerely,

Baoquan Gao.

Reviewer 2 Report (Previous Reviewer 3)

The authors have followed most of my recommendations and suggestions. I think the article should be accepted. 

If any, I miss a detailed answer to my comments indicating where the authors had introduced changes/modifications, and a more clear way to identify them in the main text (ie. using a bold font?).  Appart from that I have nothing more to suggest.

Author Response

Comments from author:
We would like to thank you for your careful reading, helpful comments, and constructive suggestions, which has significantly improved the presentation of our manuscript.

Comments:

If any, I miss a detailed answer to my comments indicating where the authors had introduced changes/modifications, and a more clear way to identify them in the main text (ie. using a bold font?).  Appart from that I have nothing more to suggest.

Response:
Thank you for your comments. I have marked the parts of the manuscript that were revised in red color, and I hope I can get your understanding. Thank you again for helping me review the manuscript

Yours sincerely,

Baoquan Gao.

This manuscript is a resubmission of an earlier submission. The following is a list of the peer review reports and author responses from that submission.

Round 1

Reviewer 1 Report

This paper discusses a problem that arose during the testing of an astronomical site: the quantitative estimation of cloud coverage using images of a full-sky camera on both moonless and moonlit nights. The method described in this article is potentially useful for other astronomers.

However, the authors make almost no mention of previous similar works, which is something that needs to be added, and they need to clearly explain how their method differs from these previous similar work. If they do so, I support the publication of the article. 

Nevertheless, the manuscript is not suitable for publication in its present form. Apart from the above mentioned and some additional minor shortcomings (e.g. missing axis labels on figures, giving not the best reference, etc., see below), the main problem is language-related. In several places the terms of the paper is unusual or even confusing. There are many typos as well. 

In order to recommend the acceptance of the manuscript, it is necessary not
only to correct the minor errors I have pointed out, but also to carry out
a careful language revision. 

==========================================================General remarks:

Capital and normal letters are often misspelled. There is often no space between words, especially after commas. In many cases, not the proper words are used. Some of these are mentioned below. Please check these and correct where necessary.

==========================================================

Title

"Nighttime cloud cover estimation for the multi-telescope system"

What does 'the' multi-telescope system mean? Please, be specific.
How about the following title, for example?
"Nighttime cloud cover estimation for the Multi-Application Survey Telescope Array (MASTA)"

1. Introduction

"Comprised of 40 optical telescopes, MASTA will separated into two parts, one will be located in the Saishiteng Mountain. The MASTA telescope has an aperture of 710 mm, and its designed optical field of view is 6 degree. When equipped with a customized large sCMOS camera, its final field of view..."

The first sentence says that the MASTA consists of 40 telescopes and a part of them (how many telescopes?) will be located in the Saishiteng Nountain. While the second sentence starts to specify a single telescope. Does this mean that 40 identical telescopes are planned to be installed? Or, are you talking about the first telescope here? Please clarify this.
You may also add a reference, if available, that describes the MASTA project.

"The method is based on the Otsu [7] dynamic threshold algorithm"

You are referring to a work that uses this method.
The original article describing the method should be cited here:
Otsu, N. (1979) IEEE Tran. Systems Man. Cyber. 9, pp. 62-66

I really miss at least one paragraph about the previous similar works. There have been studies on cloud segmentation that used the Otsu method, and even the image colorizing procedure used by the authors to improve it is well-known (see e.g. Moughyt et al. 2015, Int. Conf. Material and Energy, Proc. of ICOME 15, Paper 298; Mustaza et al. 2019, IEEE 7th Conference on Systems, Process and Control (ICSPC), pp. 102-107). Akdemir et al. (2022, Astron. Computing, 38, 100551.) discuss the same problem with similar results. These works (and potentially others) should  be mentioned here, and the extent to which the present work differs from previous ones should be highlighted.

2. Instrument and Data

2.1 Camera system

"To monitor the local sky, we installed a Starlight Xpress Oculus all-sky camera on the top of the 12-meter-high DIMM tower."

What is the DIMM tower? Please resolve the acronym.

"As shown in Fig. 1, the camera features a CCD detector with a..."

Please rephrase this sentence, or better omit it. The size of the CCD could be moved to Table 1 along with the other technical data, and the reference of Fig. 1 could be moved to the end of the next sentence, since it describes the external appearance of the equipment, which is actually shown in Fig. 1.

"The camera is controlled using the AllSkeEye software"

I would like a reference here that describes this software.

3. Method

Fig. 3

The two histograms on the right have no legend on either axis. Please add them and explain them in the caption. (e.g. ...on the horizontal axis the brightness of the pixels on a 0-255 grey scale, on the horizontal axis the number of pixels with that brightness...)

In the bottom right panel, the 'cloud' and 'sky' labels have merged. Please fix it.

3.2.1 Moon location

"The zenith angle(theta) and azimuth angle(az) of moon can b obtained Using Astropy software package"

moon -> Moon (capital letter)
can b -> can be
Using -> using (small letter)

For the sake of clarity, these angles (theta, az) and the orientation of the xy coordinate system in the plane should be defined here or referred to a work where these have been done.

Fig. 5

Axis labels are missing again as in Figure 3. Please add these.

"...Otsu is applied..."

A similar word structure is used several times later. Please use
Otsu method or Otsu's method instead. 

Fig. 7

"The black represents clear sky and the white represents clouds."

Here, the caption is not accurate, as the mask covering the Moon is also black. Please rephrase it.

4. Result and Discussion

"There are four different cloud cover for NMIs shown in Figure. 8:
clear sky, few cloudy, partly cloudy, and overcast."

few cloudy -> few clouds (everywhere, throughout the manuscript)

"...using formula 14, which are noted as experimental results."

noted -> called

"Finally, we use LabelMe software to label these images..."

Please provide a reference to the LabelMe software.

"...has an impact on the ground-truth results"

This `ground-truth result' term is very unusual. 
I'd prefer to use the terms 'exact' or 'actual', or, possibly 'real cloud cover'.

"...inaccuracy is greater when..."

greater -> higher

"The percentage of cloud cover less than 50%is 81 percent, which is nearly
identical to the results reported in ([? ])."

Apart from the missing reference, I don't understand the sentence (nor Fig. 10). What is on the vertical axis in Fig. 10? Is it the number of nights with a given cloud coverage as a percentage of the total number of nights?
This must be clarified here in the text and also in the caption of the figure.

==========================================================

Reviewer 2 Report

For the analysis of astronomical photometric observations it is important to extract the noise from the signal. The noise sources relevant to this work are clouds and the Moon, while the signals would be searched for in pixels classified as 'sky'. For that purpose, the authors have developed a numerical method to recognize cloud pixels in grey-scale images of the sky when the moon is out. The method has been adapted from the Otsu-method,
which is a segmentation method to distinguish between foreground and backgound image pixels.
That the methods works for real images, here taken near the site where the 
astronomical observations will take place, has been demonstrated in Figures 8,9, and Table 2. Overall the paper is interesting and well written. It combines several aspects of relevance to different fields of resaerch (astronomical obvservations, image processing, data reduction).
I recommend its publication in the Universe Journal after minor revision.
Throughout the text, more explanations are needed for a potential broad readership,  as outlined below.

- Abstract: some phrases should be more precise: 'for the multi-telescope system' if the authors are refering to a particular stystem ('the') which one should be mentioned, but if they refer generally to such systems, the should write more general 'for multi-telescope systems'. I guess the the authors refer here to the MASTA telsecope system?
Furthermore, 'this approach is effective and accurate' can this be quantified in terms of a recovery rate of clould pixels recognized as such by the method? Table 2 suggest an accuracy of better than 10%.

Line 26 'The MASTA telescope has' -> 'Each MASTA telescope has' (each out of the 40 telescopes has the mentionend aperture, right?)

Line 65, 'the camera features a CCD detector with a 1392 x 1040 pixel CCD detector...'  
-> the camera features a 1392 x 1040 pixel CCD detecor..

Figure 3, I do not understand the x-axis of the histograms: is it brightness of the pixels  ranging from 0 (black) to 255 (white)? Please provide this information in the Figure caption. How is 'threshold line' defined? The red line in the lower right panel appears somewhere along the rising slope.

Line 105, 'corresponding to two peaks on its histogram' I see only one peak, right 
of the red line. Where is the second peak?

Line 112-116, it remains unclear how the weights and the variances as a function of the threshold t are defined? Variance requires knowledge of a mean value for the foreground and the background images. How are those obtained? How is it known whether a pixel belongs to the foreground or the background? Or is this distinction made by means of the unknown threshold t? More explanation of the Otsu method is required.

Figures 5,6, the Moon is masked by a circle. Does this mean you method works only in case of full Moon? If Astropy is used to determine the location of the Moon on the sky, I would guess it is also possible to calculate the phase of the moon as a function of data? Is this included in your method? More explanation is required.

Figure 8, the top left image looks like an overcast case because of the grey shading. 
Otherwise, one can see stars. What is botton row and what is shown in the top row?

Line 164, 'It is clear that the segmentation precisely' no this is not so clear. 
What is the measure used to evaluate how precise the segmantation is? By eye?
The comparison to the 'ground-truth' results reveals errors of a few percent.

Line 186, 'data set collected between 4 January 2020 and 12 December 2022.'
please check the dates

Line 187-189: those lines are hard to understand, please check for language and typos

Conlusion: here, a number of abbreviations used throughout the main text should be explained again, like HSV image, NMI, MI means, Ostu, and ground-truth should be be replaced by 'by eye inspection' or whatever the authors consider appropriate

Reviewer 3 Report

Referee report on the manuscript entitled "Nighttime cloud cover
estimation for the multi-telescope system"

The manuscript describes an experimental setup to determine the cloud
cover installed at the site of the future multi-telescope
system. Using the described instrument the authors detail a new
methodology/algorithm to provide with a segmentation map that identify
the clear and cloud-covered regions in the full-sky images. In
particular, they present a new methodology to derive those
segmentation images (and the corresponding cloud cover fraction) in
bright/grey nights (i.e., nights with partial illumination by the
moon). They present a few examples of its implementation and
a comparison with by-eye/human-done labelling procedures, showing
that the procedure is as good (within an error marging) as a manual
labelling in estimating the cloud cover.

I find the new methodology very interesting, in particular as it would
automatize a procedure that cannot be performed manually in
a practical way. However, I find different problems in the current
version of the manuscript. There is a lack of references through
all the article and sometimes the description of the figures or
internal references are not propertly written. In general the English
should be improved. I am not a native English speaker, so I am very
sympathetic with language issues, but even for me there are sentences
and complete paragraph that are difficult to follow or even to understand.
Finally, there are some imprecissions that should be modified.

For all these reasons I recommend a major revision of the manuscript
prior to recommend its publication.

Detailed comments:

1) When first quoting the Otsu method, please give the proper reference:

- Nobuyuki Otsu (1979). "A threshold selection method from gray-level histograms". IEEE Trans. Sys. Man. Cyber. 9 (1): 62–66. doi:10.1109/TSMC.1979.4310076.
-  Kurita, T.; Otsu, N.; Abdelmalek, N. (October 1992). "Maximum likelihood thresholding based on population mixture models". Pattern Recognition. 25 (10): 1231–1240. doi:10.1016/0031-3203(92)90024-d. ISSN 0031-3203.

2) Sec. 1, Introduction, line 21: When quoting MASTA, please, give a reference (article?
link?). I did not find a proper reference in this regards.

3) Somewhere in the introduction, when quoting the different methods/instruments
to explore the cloud cover (lines 39-50), you should quote other systems, like, for instance:
https://ui.adsabs.harvard.edu/abs/2011PASP..123.1076A/abstract
How the procedures adopted in those instrument compare with the one presented here?

4) Sec. 2.1, Line 69: Give a reference regarding the AllSkeEye software.

5) Sec. 2.2, Data preparation, Line 81: The concept of "irrelevant
pixels" is not a good way to express the concept. What you are doing
is masking those regions in the image that are not sampling the sky,
but ground and/or structures in the ground. I think all this paragraph
is not adding too much to the overall manuscript. I would just
summarize it indicating that those pixels in the image that are not sampling
the sky but the ground or ground structures have been masked. Just a sentence.
I do not think that Figure 2 is required in this regards. It is a very simple and
straightforward procedure, and image is not required to understand it.

6) Figure 5, and indeed all maps: Please, include a colobar to all night-sky
images that allows to identify easily each greyscale with a numerical
intensity value and this way to understand more easily how the grey-scale
image corresponds to the histograms.

In the histograms I would include two labels, one with a "MI" for the top panels
and "NMI" for the bottom panels, to easily identify the images without consulting
the caption.

7) In all figures with maps, but in Fig. 4 in particular: Addopt a non-linear greyscale
color map to show the images. As they are now there is very little contrast in particular
for NMI images. In some cases, in particular when printing or visualizing in B/W systems
the lack of contrast makes the images to look totally black. Once more, include a colorbar.

8) Sec. 3.2.1: Add a reference/link to Astropy. Include the acknowledgment requested
by the astropy group.
https://www.astropy.org/acknowledging.html

9) Sec. 3.2.1: After describing equation 2 and 3, indicate the meaning of Theta.

I think that there is a bit of overdescription here. The only thing that it is done here
is that the coordinates of the Moon are transformed from cartesian to polar coordinates
(Eq. 2, and 3), and then including a distortion (Eq. 4). It is really needed this level of detail?

10) Sec. 3.2.1, Line 126: The histogram that it is shown in Fig. 5 is not an histogram of
pixels, it is an histogram of the flux values within the masked image. This is a good example
of the use of a non-formal language that I have found along all the manuscript. Please,
pay the required care and use the proper expression to indicate what it is shown in the
figures.

11) Line 127: See my comment below. The complete set of sentences from
line 127 to line 130 are confusing an not well written

12) Sec. 3.2.2: All the justification and explanation given for the colorization procedure,
include in lines from 131 to 132 is very confusing and it is not required. The claim
that "in reality, differenc colors may have (the) same luminance values...." is not mathematically correct and it is essentially not needed for the goal of this algorithm.

I would just remove all those text, and explain better the procedure. The adopted
methodology is justified as the final results presented showing the performance of the
algorithm. So, no further justificaiton is needed. Just describe what you have done, that, BTW
is not very clearly described. After reading this section several times I end-up understanding
that the procedure is the following one:

a) The grey-scale image is transformed to a three color image adopting Eq. 5,6,7 for
R, G, and B colors , by assuming that Y==Cr==Cb==MI, where MI is the original
grey-scale Moon-Image image.

b) The it is found the Max and Min values among the derive R,G, B for each pixel.

c) Then an H, S, and V parameteres are derived using Eq. 10, 11 and 12, using those Max
and Min values for each pixel.

d) Then, it is derived the f=S/V map, and finally the Otsu method is applied over this "f"
image to derive the segmentation maps segragating between clear and cloud regions in
the sky, and finally the cloud cover.

The current text include unnecesary steps. For instance, the H channel
is never used in your analyisis, and therefore, including it, and
including a full discussion/description of the HSV-color space is
irrelevant for the purposed of this exploration (and your algorithm).

I would simplify all this section and just go to the point. As a suggestion:

"We adopted a colorization procedure to generate a good proxy of the cloud cover
following the techniques decribed in [8,9]. First, a RGB image was created using
Eq. 5,6,7  (*** change Y, Cr, Cb from this eq. by MI, as you have defined Moon Images
using this nomenclature). Then, it is estimated, for each pixel, the maximun and minimum
value between the three channels of the RGB image (Eq. 8,9). Finally, we define a new image
that we will demostrate that trace well the location of the clouds as
f = (Max-Min)/Max   (** new Eq)
This is the image adopted to apply the Otsu method to estimate the cloud cover in
MIs"

Of course, this is just a suggestion, but you have included a lot of unnecesary text and
descriptions that are not needed for the goal of this study.

13) Sec. 3.3: The CCR is derived withour deprojecting the image? I
mean, it is derived on the pixel scale and not on the angular area
scale. Why? Each pixel in the image of a fish-eye system like the
adopted here does not corresponds to a fixed area in angular area
units (e.g., degree^2 or steroradian). This is strange to me. Please, justify.

14) Fig. 17: What you are showing here is not "the result" it is an example of the
use of the adopted procedure. Please, change the caption.

15) Line 163: It is the 1st time that it is presented the concept of ground-truth.
BTW, I think this is concept is not well used along this study. First, there is no way
to demonstrate the claim in Line 163: "... the results are remarkably close
to ground-truth...".  Based on what? I appreciate by-eye that there is a good
selection of the regions covered by cloud. But this is not ground-truth. Neither here
nor when used later.

I think the authors should change the concept of ground-truth by eye-selected,
eye-guided or manual labelling. In particular, remove the reference in here.
There is no basis to make this claim in this line based on Fig. 9

16) Line 167: Indicate the full number of image of the dataset to illustrate how
big is 400 images with respect to the full sample.

17) Line 171: Provide a reference and or a link to "LabelMe".

18) Sec. 4: Change the concepts of "ground-truth" by "manually labelled", and "experimental"
by "current analysis".

19) Sec. 4: The comparison presented between the manual labelling and
the currenly implemented method is a core part of this
article. However, I think it is no presented in the right way. The described experiment
does not demonstrate the absolute validity of the method or the real error. It shows
how this method compares with a manual selection of the clouds, suggesting
that indeed is as good as this other approach (within a 0.8-8.9%). Considering that the
method is automatic, its use is then justified.

However, if you want to demonstrate the real error of the method you need to
perform a different experiment. In essence you need to simulate a cloud coverage,
create synthetic images, and apply the method known the prior distribution of clouds
(input of the simulation) and compare them with the output results. I think this
could be a subjet for future explorataions and I am not suggesting that you do
such study for the current manuscript. However, I strongly suggest to change
the text. Your experiment demostrate how good is the method compared to
a manual labelling. Period. Remove the concept of ground-truth/experiment, and
change the text accordingly. 

20) Sec. 4: I do not understand the concept of "Average Error". How in detail
this parameter is derived? I think it is not clearly described.

21) Line 205: I think the experiment demonstrate that the method is
efficient and as accurate as amanual labelling, but you cannot
demonstrate that it is acurate in an absolute way.

Typos/English comments:

- Line 50:  "with the moon" -> "with illumination by the moon"

- Line 94: "has brightness characteristics similar to clouds" -> "has
  brightness characteristics similar to those produced by clouds"

- Line 103: "the values of other pixles shown follow..." -> "the
  values of the remaining pixels follow an almost normal distribution"
  (You cannot claim that it is purely normal without fitting it with a
  model, BTW)

- Line 107: "Otsu's method' -> "the Otsu method": I think the use of the Saxon Genitive
is not very formal. I would change/remove it from all the article.

- Line 112: Write the parameter "t" in italic. In Line 114 too.

- Line 112: "into to classes, and the intra-class" -> "into to classes. Then, the intra-class.."

- Line 119: "... moonlight, cloud..." -> "moonlight, and clouds..."

- Line 120: "... The cloud..." -> "Therefore, the clouds in MIS are brighter than the clear-sky areas"

- Line 122: " ... as cloud." -> "... as clouds"

- Line 123: "As result" -> "As a result"

- Line 123:" to eliminate..." -> "to mitigate its effects" (remove "of the moon")

- Line 124: "... can be calculate.." -> "... can be calculated..."

- Line 124: "... Using..." -> "using the  Astropy"

- Line 124: "... of Moon" -> "of the position (x,y) of the Moon on MIs are determined by the equations:"

- Line 124: "where(x0,y0)" -> "where (x0,y0)"

- Line 124: "we use a cubic" -> "We use a cubic"

- Line 124: "... to model the relationship..." -> "... to model this relaship.."

- Line 127: "distribution and masked" ->

"... distribution. However the masked MI have some unmasked regions
affected by moonlight.  Figure 6 shows the segmentation image obtained
when applying the Otsu method to this image.  It is clear that most of
the clouds remain undetected by this procedure. This experiment
illustrates that applying the standard Otsu method does not provide
with an optimal segmentation of clouds and clear regions. We need a new methodology
to process MIs for a better performance"

- Line 131: You repeat "colorization" and "colorize" in the same sentence.
Chane it.

- Line 131: "... are gray-scale values" -> "... corresponds to the gray-scale values of
the original image".

- Page 7, up to Line 136 -> There is a lot of English issue in this text, but I am not going
to correct them as my suggestion is to replace the full spection according to comment 12,
indlcuded above.

- Line 137: Remove the sentences from "Otsu's method..."  to "... Cloud Cover Ratios".

"Once applied the two described procedured for NMI (direct Otsu
method) and MI (Otsu method on the cloud cover proxy image, f) it is
obtained a segmentation map for each exposure (with 0 meaning clear
sky and 1 indicating the presence of clouds). This segmentation map is
the used to derive the Cloud Cover Ratios [12]"

- Fig. 7, caption: "An example of the derived cloud segmentation map based on the adopted procedure described Sec. NN for the MIs. The black color represents the regions clear of clouds
while the white color corresponds to regions covered by clouds"

- In all captions of Fig. with images: "The black color represents the regions clear of clouds
while the white color corresponds to regions covered by clouds"

- Line 143: I do not understand the meaning of the 1st sentence in Sec. 4.

- Line 150: "There are four..." -> "Figure 8 shows four examples of different NMIs showing a fully clear sky (top-panels), sky with just a few clouds (2nd row panels), partially cloudy sky (3rd row panels) and totally overcast (bottom panels), first column. A version of the same images adopting an inverse colormap is shown in the 2nd column. Finally, the resulting segmentation maps once applied the procedures described in Sec NN  are included in the 3rd column."

- Line 155: ".. no matther what sky conditions.." -> ".. no matther the sky conditions..."

- Line 158: "Figure 9..." -> "Figure 9 show a similar figure as Fig. 8, including four examples of different MIs correspoding to the same cloud coverage shon on that figure."

- Line 161: "where the black represents..." -> "using the same color scheme adopted in Fig. 8"

- Line  167: "imags" -> "images"

- Line 188: "The percentage..." -> "It is found that an 81% of the images have a cloud coverage
lower than a 50% of the full sky"

- Line 193: Change "binary image" by "segmentation image" or map. Here
  an in all the manuscript.

- Line 200: "cloud cover" -> "Cloud cover"

- Line 201: "... nighttime" -> "nightime images"

- Line 201: "and to" -> "to"

- Line 202: "Ostu" -> "The Ostu algorithm..."

- Line 202: "Four different..." -> "Four different types of cloud cover images (clear sky, low level of cloud coverage, partialy cloudy and overcast)"

- Line 208: "mainly moon area" -> "in areas with strong illumination of the Moon"